# In Silico Structure-Based Approach for Group Efficiency Estimation in Fragment-Based Drug Design Using Evaluation of Fragment Contributions

**DOI:** 10.3390/molecules27061985

**Published:** 2022-03-18

**Authors:** Dmitry A. Shulga, Nikita N. Ivanov, Vladimir A. Palyulin

**Affiliations:** Department of Chemistry, Lomonosov Moscow State University, 119991 Moscow, Russia; n4knikita@qsar.chem.msu.ru

**Keywords:** fragment-to-lead optimization, molecular modelling, de novo design, fragments, scoring function, structure-based drug design, drug design, group efficiency

## Abstract

The notion of a contribution of a specific group in an organic molecule’s property and/or activity is both common in our thinking and is still not strictly correct due to the inherent non-additivity of free energy with respect to molecular fragments composing a molecule. The fragment- based drug discovery (FBDD) approach has proven to be fruitful in addressing the above notions. The main difficulty of the FBDD, however, is in its reliance on the low throughput and expensive experimental means of determining the fragment-sized molecules binding. In this article we propose a way to enhance the throughput and availability of the FBDD methods by judiciously using an in silico means of assessing the contribution to ligand-receptor binding energy of fragments of a molecule under question using a previously developed in silico Reverse Fragment Based Drug Discovery (R-FBDD) approach. It has been shown that the proposed structure-based drug discovery (SBDD) type of approach fills in the vacant niche among the existing in silico approaches, which mainly stem from the ligand-based drug discovery (LBDD) counterparts. In order to illustrate the applicability of the approach, our work retrospectively repeats the findings of the use case of an FBDD hit-to-lead project devoted to the experimentally based determination of additive group efficiency (GE)—an analog of ligand efficiency (LE) for a group in the molecule—using the Free-Wilson (FW) decomposition. It is shown that in using our in silico approach to evaluate fragment contributions of a ligand and to estimate GE one can arrive at similar decisions as those made using the experimentally determined activity-based FW decomposition. It is also shown that the approach is rather robust to the choice of the scoring function, provided the latter demonstrates a decent scoring power. We argue that the proposed approach of in silico assessment of GE has a wider applicability domain and expect that it will be widely applicable to enhance the net throughput of drug discovery based on the FBDD paradigm.

## 1. Introduction

The language of the chemical groups determining the main reactivity and physico-chemical properties of compounds is at the heart of organic chemistry. The widespread practical use of this language implies that a significant correlation between the composition of a molecule in terms of chemical groups and the physico-chemical properties of the molecule is rather firm. This is reflected in good to excellent quantitative structure-property relationships (QSPR) obtained for many properties of organic molecules [1], e.g., for LogP [2,3], molar refractivity [4], boiling points [4,5], and others. Many of the physico-chemical properties define the ADMET [6,7,8] properties of organic molecules, thus influencing the probability of a molecule to be developed into a marketed drug.

However, the ability of a molecule to produce a desired pharmacological effect is a far more complex issue, since the inherent complexity of living systems have to be additionally accounted for. Even at a simpler level of direct in vitro ligand-receptor activity determination, the dependence of activity on fragment composition is not generally simple or even smooth. A good example is the “activity cliffs”, where a pair of molecules differing in a small group (e.g., -CH_3_) differ by an order or more in magnitude in their activity against the same target [9]. Despite this general complexity, there exist a large number of successful quantitative structure-activity relationships (QSARs) explaining the observed experimental activity in terms of chemical composition. In particular, QSAR models based on fragmental descriptors, reflecting the chemical composition of the molecule, have proven to be reliable and accurate for numerous applications [10]. Additionally, a Free-Wilson kind of QSAR modeling, in which a presence or absence of a particular group in a particular place of a general molecular structure is correlated to activity, is based on the assumption of strict additivity of each group contribution. Despite its simplicity, the Free-Wilson type of analysis is often used in practical medicinal chemistry projects to rationally guide drug development by getting insights from the data interpretation.

Particularly interesting in the studied context is the series of efforts in which physicochemical properties and activities are modeled (QSPR/QSAR) using a list of the manually curated well-defined chemical groups as descriptors, which not only afforded reasonable statistical performance but also enhanced their interpretation in terms of chemical functional groups typical for organic chemistry [11].

The general approach of statistical mechanics does not support the simple view of the additivity of contributions of the groups composing a molecule into its activity. Additional support comes from the analysis of the experimental data [12]. Despite the aforementioned, in practical settings a decent additivity is observed, at least of the first order approximation to any type of non-additive dependency. Linear models are also the simplest and hence provide the most throughput, which is often required at early stages of drug discovery when hit compounds are being sought among vast virtual libraries, and more recently even virtual chemical spaces [13]. For instance, the vast majority of scoring functions are largely additive in terms of the fragments comprising a ligand. Besides, even sophisticated computational tools in the multifactor field of drug discovery necessarily produce errors. That is why the real world drug discovery projects try to judiciously combine prediction steps with the experimental check, thus forming an Agile [14] style sequence of iterations which happen to converge sometimes to a drug, despite the fact that each iteration is guided by a rather simple incentive (an analog of anti-gradient in gradient optimization methods). Combined with the fact that the fragment contribution language for organic chemistry is natural and useful, it is no wonder that linear models are being fruitfully exploited despite the known deficiencies.

The fragment-based drug discovery (FBDD) is close to the above analysis. The core concept of FBDD, stated as a purely combinatorial problem, is that a combination of several searches done in fragment-size chemical spaces (ca. 10^7–10^) is far more efficient than a direct search for a ligand in the entire chemical space (at least 10^60^ or more) [15]. Moreover the chemical space of fragments is intentionally narrowed by requiring good ADMET properties, synthetic accessibility, and other properties. Even though FBDD is not strictly based on the additivity of the contributions of the molecular fragments, in practice a ligand is often composed of two or more fragments, each occupying a specific pocket in the binding site (“link” strategy of de novo design [16,17]). Another possibility is to grow an initially bound fragment to fill other still unoccupied pockets (“grow” strategy of de novo design [16,17]). In either case, we cannot a priori expect a significant synergy between the interactions of different fragments of a ligand and different pockets of a site. Therefore, regular FBDD optimization tends to approximately satisfy the additivity of fragment contributions.

The main source of success of FBDD is that the development is being based on the experimentally determined affinities and geometries. This minimizes the risks of poor understanding of the complex nature of binding of fragment-sized molecules to a target. Whatever a the reasons which cause a fragment to tightly bind to a certain pocket of the binding site of the target, it is safer to stem further incremental development on this information rather than on molecular modeling, for which accurate modeling of interactions of fragment-sized—in contrast to drug-like—molecules with targets is particularly challenging [18]. Another useful benefit of FBDD is that once decent strength interactions are established for a pair of fragments in different pockets of the binding site, a properly chosen linker will generally lead to a molecule combining the affinity of the two constituent fragments. One of the practical results of FBDD adoption has become the widespread use of efficiency metrics, such as e.g., ligand efficiency (LE) [19,20] and lipophilic ligand efficiency (LLE or LipLE) [21,22], which first were introduced in order to properly compare molecules of different sizes on the basis of how efficient all atoms are exploited to contribute to favorable interactions to a target. Later on, the context was broadened and the focus on the ligand efficiency has become one of the practical philosophies widely adopted in drug discovery, although with certain caveats [20,23,24]. Finally, FBDD encourages the building of ligands from the fragments with good ADMET properties in order to ensure such properties as much as possible for the drug-sized ligand.

Given the extremely time- and resource-demanding and highly multiparameter nature of drug discovery, every means to focus efforts is warranted. To this end, the knowledge regarding which part of a molecule is good enough and which part deserves further optimization is crucial to enhance productivity in drug discovery. This can be considered as a logical continuation of efficiency philosophy to the fragments composing a molecule, leading to the management of fragment contribution to the binding and fragment efficiency as analogs of ligand affinity and ligand efficiency. The main problem is how to factor the binding affinity into fragment contribution, especially taking into account its intrinsic non-additivity.

In order to address the need to estimate contributions of fragments of a molecule into its activity/affinity, several approaches have been proposed. In the realm of QSAR, the G-QSAR method suggests first to split a molecule into fragments according to fragmentation rules, defined manually for each molecule set, then to calculate descriptors for each fragment, and finally to build a QSAR model [25]. A subsequent analysis of descriptor contributions makes it possible to identify how each fragment affects the modeled activity based on the association of each descriptor to a certain fragment. Apart from descriptors solely associated with a specific fragment, the so called “interaction term” descriptors were also used in order to estimate the non-additive effect of mutual influence of several fragments on the modeled activity. The interaction term descriptors for a pair of fragments are calculated as the elements of the outer product of the vectors of descriptors used for each constituent fragment. However, despite the QSAR models with interaction term descriptors showed generally better performance than the models without them, the increase in the metrics of statistical performance is not large for the several structure-activity datasets studied. This suggests that an additive model is a good starting approximation.

The general emerging trend, referred to as a “model-structure interpretation paradigm”, in extracting structural interpretation from intrinsically hardly interpretable but highly predictive machine learning models was pointed out by Polishchuk, who reviewed several proposed methods [26]. For example, a universal approach was reported earlier by Polishchuk et al. [27] which proposes a way to single out a contribution of a molecular fragment to the predicted activity based on the predictive machine learning model regardless of its type. In this model the contribution of fragment B in molecule AB is estimated as the difference between the predictions made by the studied machine learning model made for AB and A fragments, P_AB_(B) = P(AB) − P(A). Additionally, the additivity of fragment contributions could be estimated for a specific dataset and activity at hand. It is worth noting that the fragment contributions estimated for both physicochemical (solubility) and physiological activity definitely showed a spread of values caused by different environments of the fragments in specific molecules of the dataset studied. However, the magnitude of the spread is moderate compared to the mean values of the estimated fragment contributions. This again underscores that the additive model approach can serve as a decent and useful approximation. The main disadvantage of this type of approach is that for a new structure to be meaningfully partitioned into fragment contributions, an extended set of structures with known activity should be available so as to derive the underlying QSAR model on these data.

Another approach which was proposed is based on the combination of experimental data for activity of a series of nested structure molecules and the Free-Wilson analysis made to extract the contributions of each fragment and convert them in group efficiency (GE)—an analog of LE for a certain group [28]. Thus, this approach opens the way for LE thinking based on experimental activity data for the chemically closest analogs. One of the main disadvantages of this approach is its retrospective nature. In order to obtain group contributions for a certain ligand, first a series of nested structures should be synthesized and tested for activity.

Fragment contributions to the free energy of binding can also be estimated using free energy perturbation (FEP) approaches, based on alchemical free energy (AFE) transformation and molecular dynamics- (MD) based sampling. Here the difference in free energy of binding obtained by the FEP approach between a pair of ligands is interpreted as the contribution of a group that is different between the two ligands. Substantial progress has been achieved recently in this direction using both a more elaborate but less computational resource demanding relative binding free energy (RBFE) [29,30] and a somewhat more straightforward but more resource demanding absolute binding free energy (ABFE) [31] approach. If the structure of a target is known, this approach provides the highest currently achievable accuracy in predictions of the free energy differences of binding between a pair of structures. However it comes at the cost of high computational burden and dependency on the availability of highly experienced personnel to run and interpret simulations.

The reviewed approaches to extract the contributions of specific fragments in molecules possess their benefits and deficiencies, and each method is best applicable in a certain context and on a specific drug discovery stage (Table 1). What is common between the reviewed approaches is that they mostly stem from the ligand-based drug discovery (LBDD) approach [32], whereas the relative advantages provided by the alternative structure-based drug discovery (SBDD) approach are exploited only in the FEP type of methods. However, the latter require highly skilled personnel and dedicated high end CPU/GPU resources. Recently Shulga et al. [33] proposed an approach which adheres to the structure-based drug discovery type and uses a simple setup and relies on fast scoring function-based calculations. In this approach, a contribution of each fragment of a ligand for a specific ligand-receptor complex geometry is estimated using the stakeholder based scheme, where a share of each fragment is based on the value of a scoring function for each fragment, treated as a separate molecule (capped with hydrogens or other appropriate groups, such as methyls) *at exactly the same location the fragment is situated in the whole ligand of the ligand-receptor complex* at hand. The details are given in the corresponding Methods section.

Each of the considered approaches to obtain GE values seems to be better suited at specific settings or stages of a particular drug discovery project, based on the combination of their advantages and disadvantages (Table 1). In this respect the use of the in silico R-FBDD approach fits the vacant niche of fast and lean hit-to-lead optimization, provided the receptor structure is known (Figure 1). It is well known that the accuracy of contemporary scoring functions, especially in scoring ligand-receptor poses (both scoring and ranking power), remains rather modest [34] to the extent that the manual expert curation is widely considered a good practice and a viable option [35]. Despite this the scoring functions are widely used tools at early drug discovery stages that contributed to many successful drug discovery projects. Additionally, the stakeholder scheme employed in the R-FBDD approach tends to reduce the influence of the errors in scoring functions. This is strictly true for the multiplicative errors, which cancel when a share of each fragment is calculated. Additionally, we intentionally address the issue of the influence of the scoring function choice on the results obtained from the in silico partitioning the binding affinity of a complex into the fragments’ contributions. To this end, different contemporary scoring functions are used to derive the GE values by the R-FBDD fragment contribution approach. Thus we believe the approach will find its place in the armory of tools for fruitful drug discovery. We anticipate it will be the most applicable at lean settings, for the projects lacking high cost experimental FBDD facilities or high end computational resources, both coupled with the low availability of highly skilled experts in either field.

In what follows we discuss and retrospectively check the applicability of our proposed in silico R-FBDD approach applied to the use case of a recent FBDD hit-to-lead project [36].

## 2. Experimental Based Group Efficiency Use Case

In order to outline the applicability of our proposed approach, a use case of the particular FBDD project, aimed at developing inhibitors for *Mycobacterium tuberculosis* pantothenate synthetase [36], is taken for analysis. Within this project, starting from the two fragment-sized ligands characterized by experimental *K*_D_ values and, what is even more important, resolved ligand-enzyme geometries, a further optimization step is undertaken. After a few parallel optimizations have been performed—one based on a “grow” strategy from one of the ligands and the other based on a “link” strategy—two perspective ligands with *K*_D_ values of 1.5 and 1.8 µM but somewhat suboptimal LE values of 0.28 and 0.26 kcal mol^−1^ atom^−1^, respectively, have been obtained. The generally accepted threshold value of LE for drug-like molecules is 0.30 kcal mol^−1^ atom^−1^ [20]. Thus the authors sought the ways to optimize their structures to simultaneously increase LE values and decrease *K*_D_ values closer to the nM range. The key question was to which part of the molecules one needs to focus for further optimization. The authors employed the FW-based decomposition approach based on the experimental data to find that one of the fragments and a linker have the lowest GE values (Figure 2). Perhaps the linker is amenable for synthetic considerations, so the authors focused their attention to optimization of the fragment, for which the lowest GE values were detected. The subsequent efforts resulted in a prospective inhibitor with *IC*_50_ = 253 nM and LE = 0.28 kcal mol^−1^ atom^−1^.

Our key question is whether the in silico R-FBDD approach, which we propose to estimate fragment contributions and hence GE’s, is applicable to make a decision on further optimization if it is used instead of the FW-based approach.

The two molecules **1** and **2** described above and shown in Figure 3 (corresponding to structures **5** and **8** from the work [36]) were analyzed in our work. The fragments are highlighted in different colors in the same Figure 3 according to the partitioning used in the original work [36]. In the remaining part of the work these two molecules will be partitioned into fragment contributions using the in silico fragment contribution estimation method; the contributions will be converted into GE values, and finally the thus obtained GE values will be compared with the GE values derived in Ref. [36] using the FW approach.

## 3. Methods

### 3.1. Fragment Contribution Analysis

In this section the more general Reverse Fragment-Based Drug Discovery (R-FBDD) approach [32] is briefly described, followed by its application to group efficiency (GE) estimations, and is used throughout the paper.

The main idea is based on the reasoning that within the FBDD approach we score different ligands at a binding site to rank them. It could be viewed as a process of iterating over the structure changes (basically additions) aimed at finding structures with better score and, finally, affinity/activity. This thought experiment can be considered as what we call the “forward directed” FBDD approach, in which, starting from a fragment, one iteratively finds new larger structures with better scores. However, if we change the focus inward the ligand (as we say “reverse the direction”) we arrive at the idea of the in silico Reverse Fragment-Based Drug Discovery (R-FBDD), in which we score different fragments of the whole ligand at the same site, representing them as regular molecules (with caps) in the geometries as they are in the whole ligand. Thus one is able to assess how each fragment (in the form of a small molecule) would score in the position this fragment adopts in the binding mode of the whole ligand.

Within the in silico R-FBDD approach, the fragment contributions to the binding are estimated according to the following algorithm [33].

First, the method takes on input:(a)a ligand-receptor geometry;(b)a defined set of fragments for which to estimate the contributions;(c)the interaction energy to be partitioned into contributions. It can be either experimentally determined *K*_D_, *K*_i_, −*RT* ln(*IC*_50_) or in silico evaluated *K*_D_;(d)the choice of the scoring function to use.

Second, the defined fragments are extracted from the ligand-receptor geometry and are capped with hydrogen to obtain sensible ground state molecules, preserving all the hybridization of all atoms from the ligand. Other options to cap a broken bond are also possible, such as the methyl group. It is important to note that the coordinates of fragments are not changed regarding the position of the fragment in the ligand of a ligand-receptor complex studied.

After that, for each fragment extracted and capped at previous stages, an estimation of the single point (no optimization/docking) score is done based on the value of the scoring function of interaction of each such fragment with the receptor to obtain Score(Fragment_j_) values.

Once fragment scores for all fragments are obtained, unitless fragment contributions, *ω**_j_*, are calculated according to the stakeholder Scheme (1).

Finally, once the fragment contributions are obtained, they are used to partition the interaction energy, *ΔE_mol_*, taken on input, into the energy contributions, *E_j_^Scaled^* (Equation (2)), which are by design additive to give in sum the interaction energy for the whole ligand (Equation (3)).
(1)$$\omega_{j}=\frac{Score\left(Fragment_{j}\right)}{\sum_{i=1}^{N}Score\left(Fragment_{i}\right)}$$
(2)$$E_{j}^{Scaled}=\omega_{j}\cdot \Delta E_{mol}$$
(3)$$\overset{N}{\underset{i=1}{\sum }}E_{i}^{Scaled}=\Delta E_{mol}\cdot \overset{N}{\underset{i=1}{\sum }}\omega_{i}=\Delta E_{mol}$$

The general assumptions behind in silico fragment contribution estimation analysis are:The additivity of fragment contributions in a molecule is a relevant model;An estimation of the feasibility of each fragment in a specific position of that fragment in the ligand of the ligand-receptor complex is done—not to be confused with the global search of the best binding mode of each fragment in the same binding site;A simple stakeholder scheme is adequate to partition a ligand into fragment contributions—e.g., Hirshfield analysis [37] uses the same principle to derive atomic charges from atomic centered orbitals (for also very non-additive quantity);Scoring functions are accurate enough to estimate a share; each fragment contributes to binding (estimated at a specific binding position), since the errors seem to be partially compensated—for multiplicative errors it rigorously follows from the share Equation (1).

The useful properties of the in silico fragment contribution method are:Fragment capping allows the use of different means to estimate fragment shares, ranging from different scoring functions through MM to QC calculations;Various sources of the binding affinity could be split into fragment contributions: experimental activity (transformed to energy units), scoring function at experimental geometry, scoring function at docking geometry, or other sources;No specific requirements to the ligand-target geometry exist, other than the latter is reasonable enough and is within the applicability domain of the scoring method used. Which ligand-receptor geometry (whether it is credible at all) to partition into fragment contribution is the responsibility of a researcher using the method;The method does not impose any specific requirements on how to fragment a molecule into fragments, it is the choice of a researcher, provided the fragments capped with hydrogen atoms (or other groups) at the broken bonds are sensible and within the applicability domain of the scoring function used.

When the *j*th fragment energy contributions, *E_j_^Scaled^* (Equation (2)), are obtained in the units of free energy, it is possible to convert these values into the desired group efficiencies (GE) of the corresponding fragments using the formula similar to ligand efficiency (LE) calculation (Equation (4)). In the former the affinity of the whole ligand, *ΔE_mol_*, is substituted with the corresponding estimation of fragment affinity, *E_j_^Scaled^*, whereas the number of heavy (non-hydrogen) atoms is taken separately for each fragment.
(4)$$GE_{j}=\frac{E_{j}^{Scaled}}{NH_{j}},$$
where *GE_j_*—is the group efficiency (GE) of the *j*th fragment, and *NH_j_*—is the number of heavy atoms in the *j*th fragment.

### 3.2. The Use Case Settings

#### 3.2.1. Ligand and Receptor Preparation

PDB:3IUE, PDB:3IVX were used as a source of ligand-protein geometries for structures **1** and **2** (Figure 3), corresponding to compounds **5** and **8** from Ref. [36], respectively.

In order to prepare receptors, all amino acid alternative positions, water molecules and small molecules were removed from the complexes. Hydrogen atoms were added and PDBQT files for receptors were prepared using MGLTools (v. 1.5.6) [38].

Ligands geometries were extracted from the corresponding PDB complexes. Hydrogens were added by means of OpenBabel (v. 3.0.0) [39] using the value of pH = 7.4 as a reference (with option −p 7.4). That resulted in that the carboxylic groups (fragment #4) in both ligands were deprotonated and, hence, negatively charged. PDBQT files for the ligands were prepared using MGLTools [38].

#### 3.2.2. Ligand Fragmentation Scheme

The ligands were partitioned into exactly the same fragments that were used in Ref. [36] to streamline comparison. The fragmentation is provided in Figure 3 using the similar color scheme as in the original paper.

#### 3.2.3. Ligand-Receptor Complex Geometries

##### Experimental Geometries

The first step is to estimate how the proposed approach of GE estimation works in the settings where the warnings of the ligand receptor geometry inaccuracy could be excluded. To this end, the fragment contribution analysis is done using the experimental geometries, which are fortunately available for the use case chosen for our study.

PDB:3IUE, PDB:3IVX were used as a source of ligand-protein geometries for structures **1** and **2** (Figure 3). For both ligand-receptor complexes, segments A were used among the two available segments in the crystal structures.

##### Docking Geometries

For this step all ligands, receptors and the fragmentation scheme were left untouched as in the experimental geometry case. The only difference is the source of the ligand-receptor complex geometries, which were subsequently used to derive GE values for the selected groups.

At the start of a hit-to-lead FBDD project, an experimental geometry of binding of a fragment-sized hit to the defined target is often available. In our case we use docking to find plausible binding modes for which fragment contribution analysis and subsequent GE estimation is made. Since a small molecule defined by fragment #3 (Figure 3) was previously found to be the most relevant starting position [40] for further FBDD development, so the crystal structure of this molecule, PDB:3IMC, was taken to generate plausible docking geometries for larger ligands **1** and **2**. This structure contains two similar segments, and segment A was used for further study.

Docking is known to be much better in generating native-like geometries than in properly scoring different modes [34]. Thus, in order to select the most relevant ligand-receptor geometry among the several binding modes found by docking, we use the criterion that the position of the fragment #3 (5-methoxyindole) should be the closest (in terms of RMSD) to the experimentally determined position of 5-methoxyindole bound to the same target taken from PDB:3IMC.

AutoDock Vina v1.1.2 [41] was used to generate feasible binding modes of the ligands, since it combines significant throughput and the quality of the obtained binding modes ranked as one of the best in the docking power test among the other scoring functions according to [34].

#### 3.2.4. Robustness to the Scoring Function Choice

The possible dependence of the proposed in silico approach to estimate group efficiency (GE) on the accuracy of the scoring function (SF) may be considered as its inherent weak point. However, the theoretical arguments provided in the Introduction show that (a) in the chosen niche there is no sensible alternative, and (b) the errors in the scoring function tend to partially be compensated due to the use of the stakeholder scheme to estimate the fragment’s shares. Additionally, we test our affinity partitioning approach with different SFs to estimate the variance in the predicted values and hence the ability of the approach to guide a drug discovery project in the real-life settings.

Different scoring functions were included in the experiment (Table 2). First, all types of known scoring functions—physics-based (force-field), empirical, knowledge-based and machine learning—were represented. Second, the SFs which showed decent scoring power in the CASF-2016 Update [34] experiment were preferred. Last, the choice was restricted to the publicly available SFs for our research.

The same AutoDock Vina v1.1.2 docking geometries of ligands **1** and **2** were used as a universal source of docking geometry throughout the study.

##### Settings Details

AutoDock scoring function was chosen as one of the classical publicly available physics-based (or force field-based) scoring functions. Moreover, it was shown that it performs reasonably in the context of the CASF-2013 study [46]. For AutoDock calculations, autodock4 and autogrid4 v4.2.6 were used. The grid was built using 60 × 60 × 60 points with standard 0.375 Å spacing centered about the center of mass of the ligands. All of the other options for both autogrid4 and autodock4 were standard. Option ‘epdb’ was used to run a single point calculation for the fragment-receptor geometry provided without any optimization. The input PDBQT files for both fragments and receptors were prepared using MGLTools v1.5.6 [38] as usual. The generated receptor map potentials were also used to estimate the AutoDock score as implemented within the AutoDock Vina v1.2.3 package using the ‘--scoring=ad4’ option.

AutoDock Vina was used in two versions, i.e., 1.1.2 and 1.2.3, with the same PDBQT input files as above and using the ‘--score_only’ option to produce scores.

Vinardo SF was included in the group of empirical SFs since it was reported that it appreciably improved performance above AutoDock Vina by fine tuning the settings of the latter [43]. The Vinardo SF values were calculated using its implementation within the AutoDock Vina v1.2.3 package using the option ‘--scoring=vinardo’ using the same input as for AutoDock Vina and AutoDock calculations.

DSX scoring function v0.9 was chosen as a representative among the knowledge-based type of SFs as having a reasonably high performance in scoring power among the similar type SF [44]. DSX was installed in the Anaconda environment using the ‘conda install -c insilichem drugscorex’ command.

A difficult choice was for the category of machine learning (ML) scoring functions [47,48]. Those SFs are capable of describing the non-linear dependency and take into account more details than is done in the traditional SFs. A wide range of methods are generally used in machine learning, ranging from multiple linear regression to the deep learning neural nets that were applied in the context of SF, hence spawning a great number of available options. Due to the high flexibility of the machine learning methods and the restricted set of available ligand-receptor complexes of decent quality annotated with the binding affinity, the machine learning SFs are inherently prone to overfitting [49], even though much effort to reduce this problem during the development of this type of SFs was undertaken. Interestingly, the ML approach used in one of the first ML-based type of SFs–RF-Score [50,51], the Random Forest (RF) regression, is currently employed by one of the overall best performing SF [47]. It combines the improved performance compared to the classical scoring functions and relative simplicity. In our study we chose Δ_Vina_RF_20_, also built on the RF regression over the AutoDock Vina generated features [45] to represent the ML-type of SFs, because it showed the best scoring power among the SFs tested in the CASF-2016 Update study [34]. Δ_Vina_RF_20_ was taken from the GitHub repository https://github.com/chengwang88/deltavina (accessed on 19 February 2022).

The results obtained using AutoDock Vina v1.1.2 and v1.2.3 appeared numerically close, with the difference being in the third digit (less than 1%), so only the results for v1.2.3 were used for statistical gathering. The same is applied to the pair of AutoDock v4.2.6 and AutoDock Vina v.1.2.3 with option ‘--scoring=ad4’.

## 4. Results and Discussion

For the molecules of the use case, where GE values estimated from the experimental activity data appeared to be fruitful to make an important decision on further hit-to-lead optimization, the GE values for the same fragments and molecules are obtained using an in silico R-FBDD approach. Thus the obtained GE values can be used retrospectively to choose the way of further ligand optimization. The question is therefore whether the same or similar principal decisions as were made in the work with the FW-based GE estimation could be done using our in silico counterpart.

Since the crystal structures are available for the molecules from the original work [36], we check the proposed in silico method in two settings using either: (1) the experimental geometry of the ligand-receptor complex and AutoDock Vina scoring function, and (2) the docking produced ligand-receptor complex geometry (closest to the fragments’ position) and the AutoDock Vina scoring function.

### 4.1. GE Based on Experimental Geometry

The first step of our evaluation is based on the experimental ligand-receptor complexes in order to estimate the ability of the approach to single out the group contributions. This is done in order to separate the influence of the well-known inaccuracy of the choice of the modeled (by docking) complex geometry on the prediction results.

The application of in silico fragment contribution analysis resulted in the values of fragment shares and the corresponding values of GE shown in Figure 3 designated as “R-FBDD”. The reference values of GE obtained using the FW analysis decomposition scheme based on the experimental activities of a series of nested structure ligands [36] are also provided for reference in Figure 3. It should be noted that the value of ΔG_rigid_ = 4.2 kcal/mol [52] arbitrarily added to the “scaffold” fragment #3 in Refs [36,53] was not used in our study in order to make the analysis clearer and better comparable to the corresponding Ligand Efficiency (LE) values. The obtained results are quantitatively very close. More importantly, the results agree well on the qualitative level at which the decisions on further optimization are being made.

**Figure 2 molecules-27-01985-f002:**
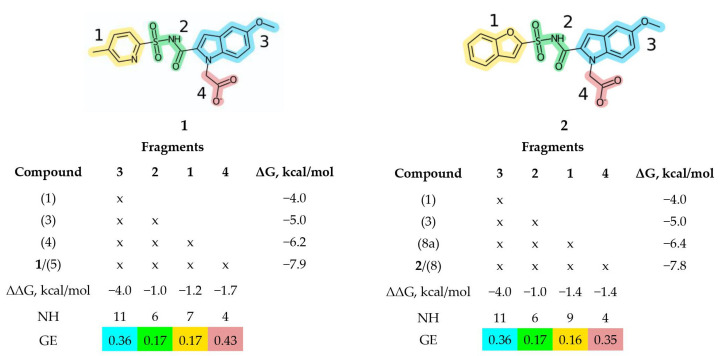
Group efficiencies for the use case compounds **1** and **2** obtained using the Free-Wilson (FW) method for a series of structure analogs, taken from Ref. [36], where compounds **1** and **2** correspond to compounds **5** and **8**, respectively. Compound numbers from Ref. [36] are in parentheses. NH is the number of heavy atoms of a fragment. The fragments of molecules **1** and **2** are numbered and highlighted with different colors. The value of ΔG_rigid_ = 4.2 kcal/mol [52] arbitrarily added to the “scaffold” fragment #3 in Refs [36,53] was not used in order to make the analysis clearer.

**Figure 3 molecules-27-01985-f003:**
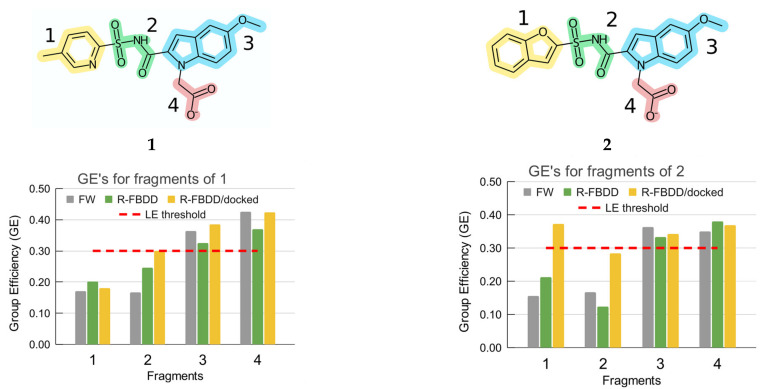
Compounds **1** and **2** corresponding to compounds **5** and **8** from Ref. [36], their decomposition into fragments and group efficiencies (GE) estimated by different methods: FW—experimental based Free-Wilson decomposition, R-FBDD—in silico decomposition based on the experimental geometry, R-FBDD/docked—in silico decomposition based on the docked geometry closest to the experimental position of fragment #3. The value of ΔG_rigid_ = 4.2 kcal/mol [52] arbitrarily added to the “scaffold” fragment #3 in Refs [36,53] was not used in order to make the analysis clearer.

One can see that the fragment contribution analysis led to quite similar results and, more importantly, it supports the same decision as was made based on the FW approach on top of the experimental activities. In particular, 5-methoxyindole (fragment #3) and acetic acid (fragment #4) exhibit very efficient binding in terms of GE, with values being above the generally accepted threshold value of LE for drug-like molecules of 0.30 kcal mol^−1^ atom^−1^ [20]. Taking into account that those fragments #3 and #4 effectively form a single scaffold, it is evident that their correct placement in the binding site guarantees an efficient “binding anchor” for a series of possible structures (see the absolute values of AutoDock Vina scoring for each fragment in Appendix A). On the other hand, fragments #1 and #2 are clearly suboptimal for the decomposition of both structures **1** and **2** according to the GE values obtained by both the experimental FW and the in silico R-FBDD approaches (Figure 3). Since the acyl sulfonamide group (fragment #2) was shown to establish crucial hydrogen bonds and to properly orient the fragment #1 into the binding pocket, the former fragment #2 was retained by the authors without further modifications. In these circumstances the most promising fragment for further optimization is fragment #1. This decision finally made by the authors of the use case is well-supported by both the reference experimental activity FW analysis and the in silico R-FBDD analysis based on the experimental geometries of the complexes of **1** and **2** with the target. Therefore, this use case demonstrates that once a decent ligand-complex geometry is available, the in silico R-FBDD analysis in terms of GE values provides a valuable basis for decision-making in lean FBDD settings.

### 4.2. GE Based on Docking Geometry

The second step strives to estimate the applicability of the proposed approach in more real-life conditions, when the exact complex geometry is not known (is too time and resource expensive to obtain). In such conditions, the GE estimation relies on the accuracy of the docking’s predicted geometries. On the other hand, such settings are far more widespread in drug discovery practice. Hence the demonstrated ability to make valuable predictions in the absence of an experimental ligand-receptor complex will mean the desired extension of availability of the FBDD methods in everyday drug discovery practice.

Among the most energetically favorable ligand-receptor complex geometries for both **1** and **2** generated by AutoDock Vina, the geometries closest in the position of the primordial 5-methoxyindole fragment (fragment #3) in its crystal structure with the enzyme were used to derive in silico fragment contributions. Notably, the resulting docking geometries do not perfectly fit the corresponding crystal geometries in the parts distant from fragment #3 (Figure 4). Despite this, the estimated GE’s for fragments of **1** and **2** based on the docking geometry (R-FBDD/docked in Figure 3) provide decent guidance for further ligand optimization.

Notably, for structure **1,** the in silico R-FBDD analysis on top of the docked ligand-receptor geometry gives quite the same results as were derived from the reference FW decomposition at the qualitative level (Figure 3 R-FBDD/docked). The docked mode used to derive the analysis is the seventh in rank but within a very close energetic range of 0.8 kcal/mol from the best pose obtained by AutoDock Vina regardless of any restraints. The GE value for fragment #1 is well below the usual LE threshold, so the main optimization focus should be placed on it according to the purely in silico analysis.

The results for structure **2**, however, deserve more attention. The mode chosen for the analysis is the second in rank and within only 0.4 kcal/mol from the best mode found. The estimation of fragment contributions and hence GE values are remarkably conserved for fragments #3 and #4. The latter is perhaps not surprising because of the RMSD filtering based on the proximity to the experimental position of fragment #3. What is different for **2** is that estimations of affinity contributions for fragments #2 and especially #1 are substantially higher than for the experimental geometry. Here the benzofuran fragment #1, according to docking, occupies a reasonable position, filling the pocket cavity and establishing a hydrogen bond with Gln72. Additionally, the energy estimate (AutoDock Vina score) for fragment #1 is considerable and even approaches that of the core fragment #3. The reference crystal structure of **2**, PDB:3IVX, contains two water molecules in this pocket, forming hydrogen bonds with Gln72 and Gln164. In addition, in the crystal structure used for docking, PDB:3IMC, which contains only fragment #3, the discussed position is occupied with a glycerol molecule whose two oxygen atoms closely overlap with the two oxygens of the water molecules discussed above. It seems that benzofuran’s moiety (fragment #1) in **2** does not fully exploit the binding possibilities offered by this pocket, which is confirmed by the remaining water molecules, the enhanced B-factor for benzofuran moiety in the crystal structure of PDB:3IVX and, remarkably for the prospective studies, by the significant variation of fragment #1 according to the positions found by docking (Figure 5 right). The latter spans a relatively narrow energy range of 1.9 kcal/mol. For purposes of comparison the similar overlay of all docking modes found for **1** is presented in Figure 5 left. Here fragment #1 does not facilitate the ligand to spread significantly between the binding modes. Interestingly, for both **1** and **2** there are a substantial number of conformations of ligands which occupy both the P1 and P2 pockets. This can additionally suggest, using only in silico means of system exploration, that fragments #1 and #2 could be optimized further in order to exploit the capabilities of the pockets more fully.

Of course, none of the approaches and tools could be expected to provide guaranteed results in such a complex and multi parameter field as drug discovery. However, a useful tool helps to get insights where the other methods do not help or are not affordable for some reason. It has been shown that even for such a non-ideal system at hand, the proposed in silico R-FBDD approach for GE estimation was capable of bringing the guidance and insights for further development in the real-life settings of the early stage FBDD.

### 4.3. Robustness to the Scoring Function Choice

Despite the expected spread of values of group efficiencies (GE) obtained with different scoring functions (SF), the values tend to be generally agreeable. Moreover, at the qualitative level, where a decision on the focus of further compound optimization is taken, different SFs lead effectively to the same decision (Figure 6).

The charged carboxylate moiety (fragment #4) receives varying scoring in physics-based and the other scoring functions. Although it establishes multiple proper salt bridges and hydrogen bonds with the receptor, their significance to the binding affinity is perhaps overestimated due to underestimated desolvation effect in AutoDock4 calculations.

The binding mode of **1**, obtained by ADV docking, generally leads to the same conclusions and decisions which would have been made using experimental geometry.

It is interesting that for the docking geometry of **2** the fragment contributions (in free energy terms) of benzofuran (fragment #1) and 5-methoxyindole (fragment #3) are consistently close numerically for different scoring functions. In the real-life settings of a lean FBDD project, this should be treated as a warning that the fundamental assumption of FBDD regarding the importance of the “anchor” (experimentally determined) fragment is violated. This may indicate that a spurious binding mode or receptor conformation might be the main cause rather than the well- appreciated affinity gain due to growing the initial fragment. In this case the R-FBDD analysis clearly shows that the binding mode chosen for analysis should not be used for further optimization, since in this binding mode fragment #1 clearly and consistently fits well into the nearby binding pocket, in part at the expense of the “anchor” fragments #3 and #4, whose GE values tend to be close to the threshold LE value of 0.3. Thus, in a real life project, another binding mode should be sought in order to base further optimization on it.

It should be pointed out also that the Δ_Vina_RF_20_ and DSX scoring functions report their score in p*K_d_* and ln(ρ_i_/ρ_ref_) units, respectively (see e.g., Appendix A). However, the nature of the stakeholder share estimation makes it possible to seamlessly use these values to partition the energy estimate into the fragment contributions.

Thus, in addition to the theoretical arguments to the expected robustness of the R-FBDD approach to GE estimation, it has been shown that scoring functions of different natures of all the known classes, despite showing appreciable numerical variation of the fragments’ affinity, result in the same decision regarding the focus of further optimization of a compound.

As practical advice, it is better to use a set of scoring functions of different types in order to estimate the variance for different fragments and obtain more robust consensus estimates.

## 5. Conclusions

The estimation of fragment contributions to the affinity of a ligand under study helps to focus the resources on its rapid and cost-effective optimization. In the context of fragment-based drug discovery (FBDD), each fragment within a ligand could be assigned a value of Group Efficiency (GE), which is a direct analog of Ligand Efficiency (LE) at a molecular level. In this work, we have briefly considered the available methods to estimate GE and group them according to different parameters, such as the reliance on ligand-based drug discovery (LBDD) or structure-based drug discovery (SBDD) approaches and the requirements to the input information that should be available to use the method. We have shown that there is a niche in this space that belongs to the early stage hit-to-lead FBDD projects which has not been filled yet.

Conceptually, we propose a new in silico SBDD method to estimate group contribution to the ligand affinity and group efficiency. We believe that this structure-based drug design (SBDD) method will complement the known existing means to estimate GE relying on a ligand-based drug design (LBDD) approach. One of the main benefits of the proposed approach is its fast and low cost in silico nature. So it helps to streamline the process of the generation of hypotheses for new active compounds at the early stages of drug discovery. Moreover, the interpretation of the fragment contributions/efficiencies in terms of ligand-receptor interactions is an additional source of insights for further structure optimization.

In order to estimate the applicability of the proposed method to real life FBDD settings, we reproduced with our approach the use case of the FBDD project in which inhibitors of the *Mycobacterium tuberculosis* pantothenate synthetase were sought. In this project the Free-Wilson (FW) method was used to partition the free energy of binding into the free energy contributions of the composing fragments, with all energies being determined experimentally for the structurally nested structures. The experimentally determined group efficiencies (GE) were used to guide further optimization of a particular fragment. We showed that by using in silico R-FBDD partitioning, which employs scoring functions, and different types of ligand-receptor geometries, that it is possible to arrive at similar decisions within the project considered retrospectively. An important practical observation is that the predicted results are relatively robust to the choice of scoring function provided the latter is capable of scoring the ligand and the fragment positions well. It is also advisable to use a set of different scoring functions to obtain more robust consensus estimates.

More experiments should definitely be conducted to estimate statistics and caveats of the method proposed applied to real life conditions to view it as a mature tool. However, even at this stage it is evident that the method will find its application in lean and agile FBDD projects that are mostly focused on early stage hit-to-lead optimization.

## Figures and Tables

**Figure 1 molecules-27-01985-f001:**
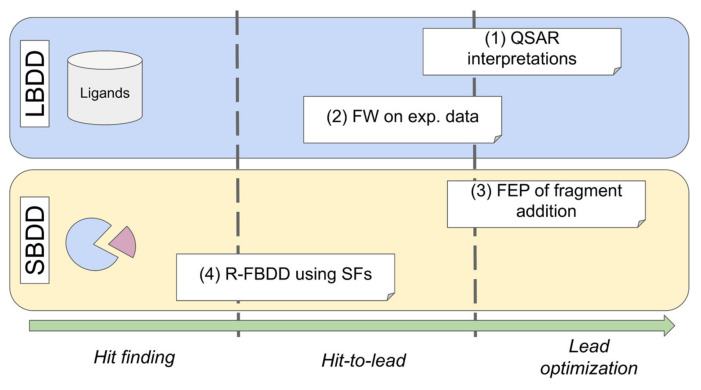
The main areas of applicability of the GE methods reviewed in Table 1 (with the corresponding methods’ numbering in parentheses) regarding a drug discovery project, using either a ligand-based (LBDD) or a structure-based (SBDD) approach. The R-FBDD fragment contribution based method fills in the vacant niche of lean in silico structure-based tools at early stages (hit finding, hit-to-lead) of drug discovery.

**Figure 4 molecules-27-01985-f004:**
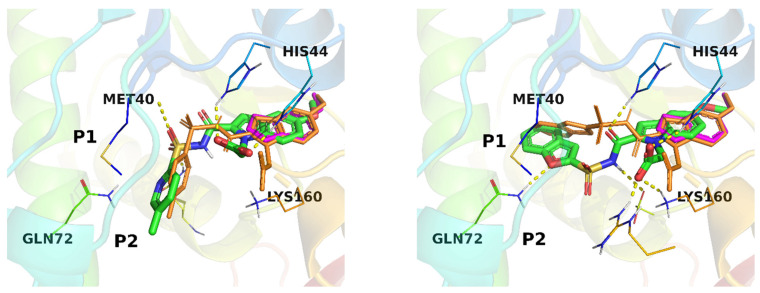
Ligand-receptor complexes for **1** (**left**) and **2** (**right**) with *Mycobacterium tuberculosis* pantothenate synthetase. The 5-methoxyindole fragment (#3) crystal position (PDB:3IMC) is in magenta. Crystal geometries of **1** (PDB:3IUE) and **2** (PDB:3IVX) are in orange. The docked geometries chosen for GE estimation are in CPK colors.

**Figure 5 molecules-27-01985-f005:**
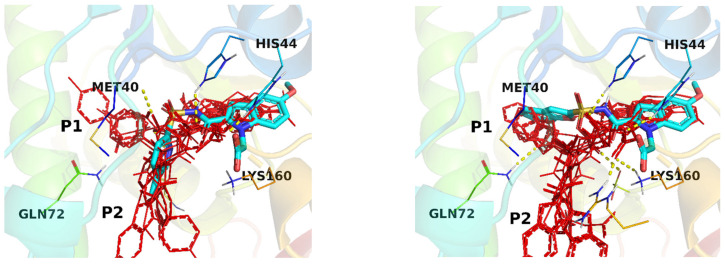
The overlay of all modes found by AutoDock Vina docking (red) for **1** (**left**) and **2** (**right**) on the corresponding crystal structures of the same molecules.

**Figure 6 molecules-27-01985-f006:**
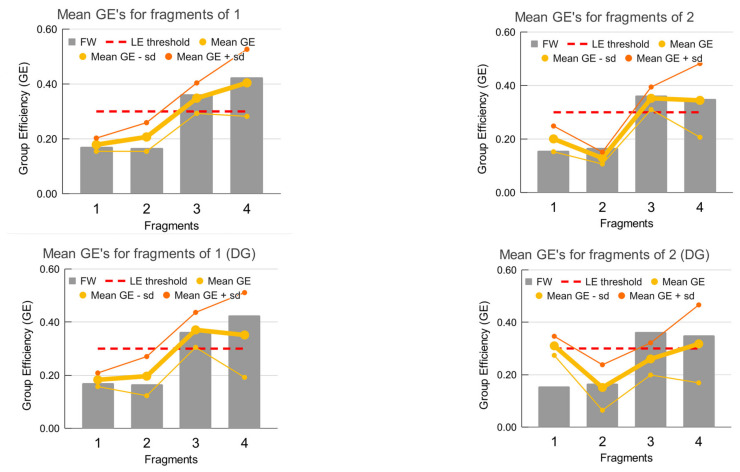
Comparison of GE values obtained using different scoring functions with the FW experimental affinity decomposition results: top—using the experimental geometries, PDB:3IUE for **1** and PDB:3IVX for **2**, and bottom—using the docking geometry (ADV) closest by RMSD to the experimental position of fragment #3 (PDB:3IMC).

**Table 1 molecules-27-01985-t001:** Pros, cons and applicability domain of different fragment contribution methods.

Method	(1) G-QSAR, QSAR Interpretation Methods	(2) Experimental Affinity Free-Wilson Partitioning	(3) FEP Estimation of Fragment Addition	(4) Fragment Contribution Based on Scoring Function
Essence	Extract contribution of each fragment via statistical analysis of its influence on the predicted activity	Extract fragment contribution from a minimal set of structurally nested ligands with known experimental activity	The simulated free energy differences between a pair of ligands are used to estimate the contribution brought by the changed group	Extract fragment contribution from a scoring function estimation of each fragment contribution, provided a relevant ligand-receptor complex geometry is available
Pros	The most robust and statistically significant Non-additive dependencies can be studied Ligand-based, no need to rely on receptor structure Fragment contributions can be calculated prospectively for new structures but within the applicability domain of the underlying QSAR models	The least possible efforts can be made in terms of synthesis and activity measurement of a small series of structural analogs The activities of the structures are valuable by themselves and can be rationalized even without building a model—by a qualitative SAR analysis	Potentially takes into account many dynamic and statistical effects associated with binding Only physics-based models (force fields or QM/MM) are used—no reliance on any additional mathematical models Additional statistical metrics from the MD trajectory regarding ligand-receptor interactions No reliance on any chemical class	“Cold start”—no need to synthetize and measure the experimental activity of any structure Several feasible binding modes can be considered for further ligand optimization Often clear interpretation of the nature of the fragment-receptor interactions No dependence on congeneric series of ligand—any chemical structure suits
Cons	An extended series of ligands with established activity is needed to ensure statistically significant results New structures, for which partitioning into fragment contribution is made, should be within the applicability domain of the underlying QSAR models	Partitioning of affinity in fragment contribution is done *retrospectively*, i.e., only *after* the structures are synthetized and their activity is measured	A steep learning curve for beginners in the field due to the inherent complexity Dedicated computational resources, often requiring high end CPU/GPU	The relevant geometry of the target needs to be known Reliance on the geometry of ligand-receptor complex Moderate accuracy of scoring functions Dependence of activity estimation on structural choices: account of crystallization waters, protonation states of receptor’s residues and a ligand, etc.
Applicability	The most applicable to lead optimization, when an extended series of different structures with known experimental activity is known	The most applicable to the hit-to-lead stage, where a fragment-sized hit is expanded to a drug-sized molecule. The FW scheme should be used to plan the synthesis of a minimal series of analogs	The most applicable to the lead optimization stage, where the chemical modifications are generally small and the binding mode is well established. Can be applied at hit-to-lead stage also by the experienced teams	The most applicable at early stages of hit finding and/or hit-to-lead, when a reliable/relevant structure/conformation of the target is known.
Principle	Ligand-based drug design (LBDD)	Ligand-based drug design (LBDD)	Structure-based drug design (SBDD)	Structure-based drug design (SBDD)

**Table 2 molecules-27-01985-t002:** Scoring functions used in the research.

Scoring Function	Source	Classification	References
AutoDock4	AutoDock v4.2.6	Physics-based function	[38]
AutoDock4	AutoDock Vina v1.2.3	Physics-based function	[38,42]
AutoDock Vina	AutoDock Vina v1.1.2	Empirical scoring function	[41]
AutoDock Vina	AutoDock Vina v1.2.3	Empirical scoring function	[42]
Vinardo	AutoDock Vina v1.2.3	Empirical scoring function	[42,43]
DSX	DrugScoreX v0.9	Knowledge-based potential	[44]
Δ_Vina_RF_20_	https://github.com/chengwang88/deltavina (accessed on 19 February 2022)	Machine-learning model	[45]

## Data Availability

The main data presented in this study is contained within the Appendix A to this article.

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
