# Peer review of "In Silico Structure-Based Approach for Group Efficiency Estimation in Fragment-Based Drug Design Using Evaluation of Fragment Contributions"

_molecules, 2022, doi:10.3390/molecules27061985_

Round 1

Reviewer 1 Report

This is well written paper advocating the fragment based drug discovery (FBDD) approach for drugs design.

Authors demostrated their promising theoretical  approach on the Mycobacterium tuberculosis pantothenate synthetase test-case.

Although more computational work is to be done to estimate statistics of the proposed method in various real life conditions, even at this stage it is evident that the method will find its application.

Publish as such.

Author Response

Response to Reviewer 1 comments

Point 1: …more computational work is to be done to estimate statistics of the proposed method in various real life conditions...

Response 1: Agree that it's needed. We plan to develop and apply the method further. In this manuscript we added a section with application of different scoring functions to the same test-case in order to imitate different real life conditions and to estimate the robustness of the method to the choice of the scoring function. We also believe that the analysis provided in the Introduction shows that in the niche of lean SBDD projects the proposed approach is highly competitive even taking into account its shortcomings.

Reviewer 2 Report

A well-documented research paper with an extended introduction.

The main problem is that the introduction is extremely extended, and the rest of sections are too short by comparing.  Therefore, is mandatory that authors extend the all the others sections of the article.

Observations:

3.2. The use case settings – links from Ref. 38 and Ref. 39 redirect to the same site: https://doi.org/10.1186/1758-2946-3-33

Line 351 – please, avoid this type of statements “best docking power”

Figure 2 – “The docked geometries chosen for GE estimation are in color.”???

 Maybe “The docked geometries chosen for GE estimation are in CPK colors.”

Author Response

Response to Reviewer 2 comments

Point 1: The main problem is that the introduction is extremely extended, and the rest of sections are too short by comparing. Therefore, is mandatory that authors extend the all the others sections of the article.

Response 1: 1. We believe the Introduction helps to theoretically place the proposed method into the context of Group efficiency (GE) application. That's why it reviews the existing types of approaches and compares them.
2. As also the other reviewers suggested, we elaborated the method section to be more clear and readable and added a section to the results and  discussion, where different contemporary scoring functions are compared on the same test case. We hope that would result in a more balanced manuscript text.

Point 2: 3.2. The use case settings – links from Ref. 38 and Ref. 39 redirect to the same site: https://doi.org/10.1186/1758-2946-3-33

Response 2: Fixed.

Point 3: Line 351 – please, avoid this type of statements “best docking power”

Response 3: Fixed.

Point 4: Figure 2 – “The docked geometries chosen for GE estimation are in color.”??? Maybe “The docked geometries chosen for GE estimation are in CPK colors.”

Response 4: Fixed.

Reviewer 3 Report

The authors developed an in-silico approach by assessing GE of a decomposed molecule. They aim to achieve a broad applicability domain and thereby enhance the throughput of FBDD in real life. The new method is promising and may contribute to practical molecular design. However, the manuscript needs significant improvement before publishing on MOLECULES. 

Major comments:
1.  The method section is difficult to follow. Lots of details should be included. For example, instead of roughly listing the steps of “On input”, the authors may use complete sentences to describe and provide more details. The authors may consider drawing a flowchart to demonstrate their protocol. 
2. The authors should significantly improve the readability of the method section. It is full of jargon, e.g., “forward directed”, “reverse the direction”, R-FBDD, which is not friendly for general audiences in the field. The authors may give concise but precise descriptions when they mention them first rather than just cite corresponding papers. 
3. The authors only use one example to demonstrate their new method. A single case is not convincing. The authors may find more cases to show the new method’s usefulness. 
4. I found many hard-to-read sentences in the “Method and Results and discussion” sections. The authors should very carefully polish them. 

Minor comments:
1. Compounds 1 and 2 do not contain any chiral centers. Why did the authors draw the 2D structures with a chiral notation (sulfonamide)? 
2. The structure 3UIE does not contain compound 1. Please provide the correct ID. 

Author Response

Response to Reviewer 3 comments

Point 1: The method section is difficult to follow. Lots of details should be included. For example, instead of roughly listing the steps of “On input”, the authors may use complete sentences to describe and provide more details. The authors may consider drawing a flowchart to demonstrate their protocol.

Response 1: Agree, we extended the description.

Point 2: The authors should significantly improve the readability of the method section. It is full of jargon, e.g., “forward directed”, “reverse the direction”, R-FBDD, which is not friendly for general audiences in the field. 
The authors may give concise but precise descriptions when they mention them first rather than just cite corresponding papers.

Response 2: Agree, we improved the text.

Point 3: The authors only use one example to demonstrate their new method. A single case is not convincing. The authors may find more cases to show the new method’s usefulness.

Response 3: We agree that multiple cases would make the paper even more convincing. However, we believe that:
1. A direct comparison with the work, where GE was used to guide the development is crucial. Such papers are extremely rare;
2. We extensively reviewed the existing literature to extract the theoretical grounds used to estimate GE. On that basis we show that, whatever accurate, the proposed approach fills in the niche that was not covered by the other methods;
3. We decided to consider in detail one case and in order to better estimate the performance of the method in the real life conditions we added  estimations based on different types of modern scoring functions. We hope it'll answer some questions and increase credibility.

Point 4: I found many hard-to-read sentences in the “Method and Results and discussion” sections. The authors should very carefully polish them.

Response 4: Agree, refactored.

Point 5: Compounds 1 and 2 do not contain any chiral centers. Why did the authors draw the 2D structures with a chiral notation (sulfonamide)?

Response 5: Agree, fixed.

Point 6: The structure 3UIE does not contain compound 1. Please provide the correct ID.

Response 6: Agree, fixed.

Reviewer 4 Report

In the framework of Fragment Based Drug Design (FBDD), the authors, tried to expand the applicability of their in silico Reverse Fragment Based Drug Discovery (R-FBDD) methodology previously published in Mendeleev Communications [2021, 32, 291-293]. Thus, the novelty of the method is not new but its retrospective application on the data from Ciulli et al. [Hung AW, Silvestre HL, Wen S, George GP, Boland J, Blundell TL, Ciulli A, Abell C. Optimization of Inhibitors of Mycobacterium tuberculosis Pantothenate Synthetase Based on Group Efficiency Analysis. ChemMedChem. 2016,11, 38-42] is well addressed. In order to well validate this R-FBDD method more fragments / molecules should be analyzed as the authors themselves also stated in the last sentences of the conclusions. Furthermore, some experimental parts of the work should be deeply revised. The main issue lies in the use of Autodock Vina's scoring function (version 1.1.2) to estimate the group efficiency (GE). Indeed, it is well known that Autodock Vina’s docking engine is highly capable to reproduce binding poses, but the empirical nature of the scoring function is far from usable for this study, too approximate. Instead, authors can use and / or compare Autodock4's scoring function which is force field based and therefore more suitable. Of note, this scoring function (ad4) was also recently introduced in the new version of Autodock Vina [Eberhardt J, Santos-Martins D, Tillack AF, Forli S. AutoDock Vina 1.2.0: New Docking Methods, Expanded Force Field, and Python Bindings. J Chem Inf Model. 2021, 61, 3891-3898]. A further methodological hint could be the employment of a rescoring method which improves the accuracy of the estimated binding affinities [Li H, Leung KS, Wong MH, Ballester PJ. Improving AutoDock Vina Using Random Forest: The Growing Accuracy of Binding Affinity Prediction by the Effective Exploitation of Larger Data Sets. Mol Inform. 2015, 115-126].

Minor revisions and typos. The introduction section of the manuscript is too long, that length is more suitable for a review rather than for an article. The reference numbers on the square brackets should be not in superscript style. In the references section, few of them are not formatted in the right style. Page 8, line 347: 5-methoxyindole is written in bold style. Page 13: the TOC figure seems to have a very low resolution.

Author Response

Response to Reviewer 4 comments

Point 1: In the framework of Fragment Based Drug Design (FBDD), the authors, tried to expand the applicability of their in silico Reverse Fragment Based Drug Discovery (R-FBDD) methodology previously published in Mendeleev Communications [2021, 32, 291-293]. Thus, the novelty of the method is not new but its retrospective application on the data from Ciulli et al. [Hung AW, Silvestre HL, Wen S, George GP, Boland J, Blundell TL, Ciulli A, Abell C. Optimization of Inhibitors of Mycobacterium tuberculosis Pantothenate Synthetase Based on Group Efficiency Analysis. ChemMedChem. 2016,11, 38-42] is well addressed.

Response 1: We found that the more general R-FBDD approach to estimate fragment contributions is well suited to the Group efficiency (GE) context and thus decided to explore the field deeper. We reviewed the current literature on GE and found that despite the GE term being as appealing as LE, the estimation of the former is far harder (incurs different approximations). We also revealed that different existing approaches have their pros and cons and are best applicable to different circumstances/stages of drug discovery. We also found theoretically that our proposed approach for GE estimation has its own unique niche that has not been properly covered by the other methods. So we liked to highlight our theoretical findings and support it with a case study to see it in action.

Point 2: In order to well validate this R-FBDD method more fragments / molecules should be analyzed as the authors themselves also stated in the last sentences of the conclusions.

Response 2: Agree, but we feel that it would be out of the scope of the  present paper. We plan to do it in the near future in separate, more focused papers.

Point 3: Furthermore, some experimental parts of the work should be deeply revised. The main issue lies in the use of Autodock Vina's scoring function (version 1.1.2) to estimate the group efficiency (GE). Indeed, it is well known that Autodock Vina’s docking engine is highly capable to reproduce binding
poses, but the empirical nature of the scoring function is far from usable for this study, too approximate. Instead, authors can use and / or compare Autodock4's scoring function which is force field based and therefore more suitable. Of note, this scoring function (ad4) was also recently introduced in the new version of Autodock Vina [Eberhardt J, Santos-Martins D, Tillack AF, Forli S. AutoDock Vina 1.2.0: New Docking Methods, Expanded Force Field, and Python Bindings. J Chem Inf Model. 2021, 61, 3891-3898].
A further methodological hint could be the employment of a rescoring method which improves the accuracy of the estimated binding affinities [Li H, Leung KS, Wong MH, Ballester PJ. Improving AutoDock Vina Using Random Forest: The Growing Accuracy of Binding Affinity Prediction by the Effective Exploitation of Larger Data Sets. Mol Inform. 2015, 115-126].

Response 3: Thank you for the great advice! We initially didn't like to focus on testing different scoring functions for the sake of the manuscript brevity and clarity, despite the fact that we had preliminary done such kind of work.
We decided to add a specific section, which compares how the choice of a scoring function affects the predicted results. We included the proposed AD4 physics based scoring function. As for ML-based SF, for the moment they are numerous and it is out of scope of the paper to compare them and select
the most suited one. We decided to take a conservative position and used DVinaRF20 scoring function to the benchmark, since it was considered in the latest available CASF-2016 Update study, where this SF showed the highest scoring power among the other SFs in the study. We also added a
knowledge-based DSX SF to finally cover all the types of SFs.
We agree that this new section will help to outline the expected performance in the real life settings (with the applicability domain) and finally make the proposed method more credible.

Point 4: The introduction section of the manuscript is too long, that length is more suitable for a review rather than for an article.

Response 4: Yes, this is essentially a mini-review in the field of GE. We believe it's necessary to establish theoretically that the proposed method has its benefits in the niche, where the other methods are hardly applicable at all.
So it's the meaningful part of our message. And the use case demonstrates the approach in detail.

Point 5: The reference numbers on the square brackets should be not in superscript style. In the references section, few of them are not formatted in the right style.

Response 5: Agree, fixed.

Point 6: Page 8, line 347: 5-methoxyindole is written in bold style.

Response 6: Agree, fixed.

Point 7: Page 13: The TOC figure seems to have a very low resolution.

Response 7: Agree, fixed.

Round 2

Reviewer 2 Report

The authors successfully responded to all my observation/requests, therefore I consider that the paper is fitted for publication in Molecules. 

Author Response

Thank you.

Reviewer 3 Report

The authors have significantly improved the manuscript. MOLECULES may consider publishing this work after the authors' further writing improvement. 

Author Response

Response to Reviewer 3 comments

Point: MOLECULES may consider publishing this work after the authors' further writing improvement. 

Response: We made an additional improvement of writing of the manuscript.

Reviewer 4 Report

The authors responded fairly well to the review requests and the manuscript is improved. However, the introductory part remains too long. It is understandable that this part is important to highlight the author's research background but it can be further summarized by using schemes or tables. Molecules, after minor revisions and writing improvements, may consider publishing this manuscript.

Author Response

Response to Reviewer 4 comments

Point 1:  However, the introductory part remains too long. It is understandable that this part is important to highlight the author's research background but it can be further summarized by using schemes or tables.

Response 1: Thank you for the advice. We inserted two new figures. One summarizes the main differences of the existing methods of estimating GE and hence complements the summary table at a higher level. Another figure describes the reference use case of Free-Wilson experiment and makes it more comprehensible than a pure text representation.

Point 2: Molecules, after minor revisions and writing improvements, may consider publishing this manuscript.

Response 2: We made an additional improvement of writing of the manuscript.